# Exploring an Alternative Configuration of the Hydroclimatic Modeling Chain, Based on the Notion of Asynchronous Objective Functions

**Simon Ricard** [1,*] , **Jean-Daniel Sylvain** [2] **and François Anctil** [1]

1    Department of Civil and Water Engineering, Laval University, Québec, QC G1V 0A6, Canada; Francois.Anctil@gci.ulaval.ca
2    Direction de la recherche forestière, Ministère des Forêts, de la Faune et des Parcs, Québec, QC G1P 3W8, Canada; jean-daniel.sylvain@mffp.gouv.qc.ca
\*    Correspondence: simon.ricard.1@ulaval.ca

**Abstract:** This study explores an alternative configuration of the hydroclimatic modeling chain around the notion of asynchronous objective-function (AOF). AOFs are calibration criteria purposely ignoring the correlation between observed and simulated variables. Within the suggested alternative configuration, the hydrologic model is being forced and calibrated with bias corrected climate variables over the reference period instead of historical meteorological observations. Consequently, the alternative configuration circumvent the redundant usage of climate observation operated within conventional configurations for statistical post-processing of simulated climate variables and calibration of the hydrologic model. AOFs optimize statistical properties of hydroclimatic projections, preserving the sequence of events imbedded within the forcing climate model. Both conventional and alternative configurations of the hydroclimatic modeling chain are implemented over a mid-size nivo-pluvial catchment located in the Saint-Lawrence Valley, Canada. The WaSiM-ETH hydrological model is forced with a bias-corrected member of the Canadian Regional Climate Model Large Ensemble (CRCM5-LE). Five AOFs are designed and compared to the common Kling-Gupta efficiency (KGE) metric. Forced with observations, AOFs tend to provide a hydrologic response comparable to KGE during the nival season and moderately degraded during the pluvial season. Using AOFs, the alternative configuration of the hydroclimatic modeling chain provides more coherent hydrologic projections relative to a conventional configuration.

**Keywords:** hydroclimatology; modeling chain; objective functions; catchment scale

---

## 1. Introduction

Many studies assess the impact of climate change on regional water flow regimes by implementing a hydroclimatic modeling chain [1–6] that translates climate variables projected by Global Climate Models (GCM) or Regional Climate Models (RCM) into the future hydrologic regime of a given watershed. Conventional configurations of the hydroclimatic chain (Figure 1) first apply statistical post-processing to simulated climate variables in order to minimize mismatches with observations. Quantile mapping [7,8] is a common post-processing method which defines transfer functions that relate empirical distributions of climate observations and simulations over an overlapping reference period. Corrected climate variables over reference and future periods are then produced applying the transfer function to raw simulations. In parallel, a hydrologic model is forced with climate observations, simulating hydrologic processes at the catchment scale. Through an iterative process, an optimization algorithm calibrates the free parameters of the hydrologic model according to a given objective-function:

A criterion minimising the error between simulated and observed streamflow. Hydrologic projections over reference and future periods are finally produced forcing the calibrated hydrologic model with the corrected climate variables.

Albeit frequently used to assess the impact of climate change on water resources, conventional configurations of hydroclimatic chains raise concerns regarding their ability in producing consistent hydrologic projections. These latter operate quantile mapping and calibration independently without ensuring consistency between the redundant usages of climate observations (dashed lines, Figure 1). Climate data heterogeneity and scarcity are among the most important limitations of hydroclimatic modeling [9–11], motivating the use of modeling chains that rely exclusively on air temperature and precipitation. Surrogating the deficient observation coverage by biased reanalyses or remote sensing products often corrode the resulting simulated hydrological response [12,13]. In this context, avoiding a non-added-value redundant use of climate observation may potentially lead to a reduction of the overall uncertainty affecting the hydroclimatic modeling chain [14,15].

In opposition to meteorological applications, climate modeling is not constantly updated in order to better reproduce the observed conditions of the climate system, these latter being exclusively imposed at the onset of the projection run. It has for consequence that simulated climate series rapidly depart from the sequence of observed climate events over the historical period (meteorology) but the statistical properties of the climate system is preserved over a few decades [16]. Applied to simulated climate time series, quantile mapping conducts an asynchronous transformation which preserves the sequence of events embedded within the climate model [17]. In contrast, hydrologic models are typically trained in reproducing the sequence of events observed over the historical period. Since most climate change impact studies on water resources assess the projected change in statistical properties between a reference and a future simulated flow regime [18], the use of the correlation component in the objective functions appears questionable. Does it bring added-value to the resulting hydrologic projections? Or on the other hand, does it taints the parametric identity of the calibrated model in a way that would corrode the credibility of the resulting projections [19]? The redundant use of climate observations may also be circumvented using bottom-up vulnerability-based (scenario-free) approaches assessing the impact of climate changes on water resources [20–22].

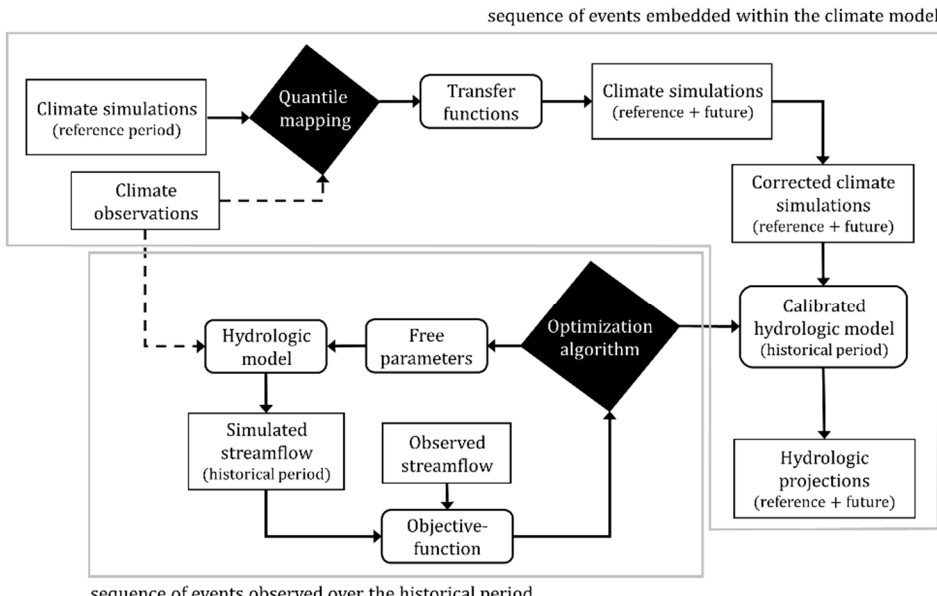

**Figure 1.** A conventional configuration of a hydroclimatic modeling chain.

In this study, we propose an alternative configuration of the hydroclimatic modeling chain that forces and calibrates the hydrologic model directly with post-processed climate simulations, instead of observations (Figure 2). Since the sequence of events embedded within the climate model differs

from historical observations, a calibration criteria must purposely ignore correlation between observed and simulated streamflow. Such a criteria is hereafter referred to as "asynchronous objective-function" (AOF, defined in Section 2.1). We first evaluate in Section 3.1 the hydrological performance of five exploratory AOFs forcing the WaSiM-ETH hydrologic model with climate observations over three mid-size catchments located in the St. Lawrence Valley, Canada (Section 2.3). We subsequently examine the capacity of the alternative configuration in constructing consistent hydrologic projections over the reference period simulated by the climate model (Section 3.2). We expect correlation-based calibration to dominate AOFs over the observed historical period, whereas AOFs provide more accurate hydrologic projections. The scope of the study remains a proof a concept aiming to define the notion of asynchronous objective functions (AOFs) and demonstrating its applicability in the scope of climate change impact studies.

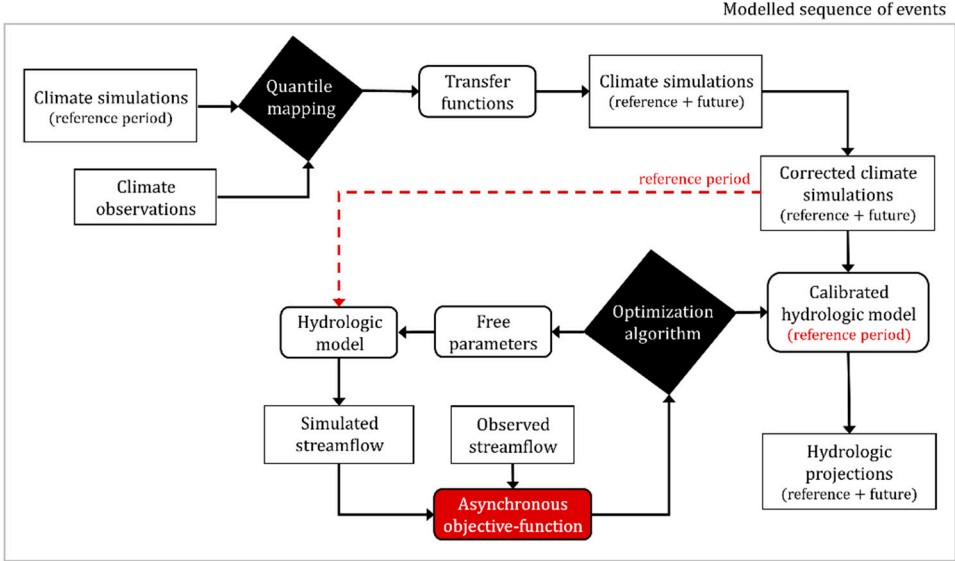

**Figure 2.** Alternative configuration of the hydroclimatic modeling chain.

## 2. Methodology

The proposed step-by-step methodological framework (detailed in Appendix A) aims firstly at the testing of five exploratory AOFs in a standard modeling framework: The hydrologic model is forced with climate observations and performance is evaluated between synchronized observed and simulated streamflow values. AOFs are compared to the KGE metric over three catchments using a common split-sample test. Any AOFs presenting inappropriate performance will then be excluded from further analysis. A site-to-site variability of the simulated hydrological response is performed next to confirm the good behavior of the modeling chain before it full application to a single site. Hydrological projections are subsequently constructed from climate model simulations. Since projections are not synchronized to observations, performance is evaluated using statistical criteria excluding correlation. The evaluation of the split sample-sample test (calibration/validation periods) remains consistent with previous analyses.

### 2.1. Asynchronous Objective Functions

Commonly referred to as 'calibration metrics' or 'optimization metrics', objective functions are goodness-of-fit measures orienting a calibration process toward an optimal parametric solution. The numerous calibration metrics described in the literature [23] are used individually or in a group of two or three [24–26]. It is also possible to transform the streamflow time series prior to using a metric in order to attribute more weights to high, intermediate, or low flows [27]. The selection of a given objective-function is known to affect the response simulated by a hydrologic model forced with either

climate observations [28,29] or climate model simulations [27,30,31]. Most common calibration metrics, among which the root mean square deviation (RMSD, [32]), Nash–Sutcliffe efficiency (NSE, [33]) and Kling–Gupta efficiency (KGE, [34]), contain a correlation component, which means they are designed to provide an optimized simulated time series synchronized with observations.

Asynchronous objective functions (AOFs) are here defined as calibration metrics that purposely neglects to account for the correlation between the observed and simulated variables. AOFs rather minimize deviations of other statistical properties between the observed and simulated variables. Five exploratory AOFs are described in Table 1 and tested in Section 3. *AOF1* refers to the root mean square deviation applied to the interannual hydrograph (Equation (1)):

$$AOF1 = \sqrt{\frac{\sum_{i=1}^{365}\left(\overline{Q_{obs,i}} - \overline{Q_{sim,i}}\right)^2}{365}} \tag{1}$$

where $\overline{Q_{obs,i}}$ and $\overline{Q_{sim,i}}$ are observed and simulated mean annual streamflow, and $i$, the day of the year.

*AOF1* relies on the assumption that the hydrologic regime associated to the climate model should be imprinted with statistical properties comparable to observations, notwithstanding the distinct sequence of daily events. *AOF2* seeks minimizing the absolute deviation (*AD*) in the $n$-th moments of the observed and simulated streamflow distributions (Equation (2)):

$$AOF2 \;=\; AD_{n,t} = \left|\mu_{n,t}^{sim} - \mu_{n,t}^{obs}\right| \tag{2}$$

*AOF3* refers to the absolute deviation (*AD*) between mean values ($\mu_1$) for the $m$-quantiles of the simulated and observed distributions (Equation (3)):

$$AOF3 \;=\; AD_m = \left|\mu_{1,m}^{sim} - \mu_{1,m}^{obs}\right| \tag{3}$$

*AOF4* and *AOF5* (Equations (4) and (5)) result from the combinations of *AOF1* with *AOF2* and *AOF3*, respectively:

$$AOF4 = [AOF1; AOF2] \tag{4}$$

$$AOF5 = [AOF1; AOF3] \tag{5}$$

*AOF1* is the most straightforward AOF since it is constructed through a single optimisation criteria and does not require a Pareto-based optimisation algorithm in opposition to others AOFs. *AOF2* is configured to optimise the first three moments: Mean, variance and skewness ($n$ = 1 to 3, Table 1). A biannual sub-scaling ($t$ = 2) preprocessing is applied, so the moments are optimised distinctly for the nival (December to May, DJFMAM) and pluvial seasons (June to November, JJASON, see Section 2.3 for a description of the hydrologic regime). The resulting number of optimisation criteria for *AOF2* thus reaches 6. *AOF3* is configured to optimise the mean values of five quantiles from the streamflow distributions ($m$ = 1–5 without any temporal sub-scaling, thus five optimisation criteria). Since *AOF1* is equivalent to a first order criteria, we excluded the first moment ($n$ = 1) from *AOF4* and the 50th percentile value ($m$ = 3) from *AOF5* to avoid the potential redundancy.

**Table 1.** Description and configuration of asynchronous objective functions (AOFs).

|  | Description | Equation | Configuration | Number of Criteria |
|---|---|---|---|---|
| *AOF1* | Root mean square deviation (RMSD) between observed ($\overline{Q_{obs,i}}$) and simulated mean annual streamflow ($\overline{Q_{sim,i}}$), where *i* is the day of the year. | (1) | - | 1 |
| *AOF2* | Absolute deviation (AD) of *n*-th moments ($\mu_n$) with temporal sub-scaling (t) | (2) | $n = 1–3$ $t = 2$ | 6 |
| *AOF3* | Absolute deviation (AD) of the *m*-th quantiles mean values ($\mu_1$) | (3) | $m = 1$ to 5 | 5 |
| *AOF4* | Combination of *AOF1* and *AOF2* | (4) | $n = 2, 3$ $t = 2$ | 5 |
| *AOF5* | Combination of *AOF1* and *AOF3* | (5) | $m = 1, 2, 4, 5$ | 5 |

*2.2. Alternative Configuration of the Hydroclimatic Modeling Chain*

Figure 2 (see Introduction) depicts the proposed alternative configuration of the hydroclimatic modeling chain, which aims to circumvent the typical redundant usage of the climate observations that affect conventional configurations (Figure 1). Quantile mapping of simulated climate variables is operated identically to the conventional configuration. The hydrologic model is forced however with corrected climate variables over the reference period and calibrated according to a given asynchronous objective-function (AOF, Section 2.1). The latter minimises statistical deviations between simulated hydrologic projections over the reference period and their corresponding observations. Hydrologic projections are produced by forcing the hydrologic model calibrated over the modeled reference period with corrected climate variables. In opposition to the conventional configuration of the hydroclimatic modeling chain, the alternative configuration do only requisite a single usage of the climate observations, it is fully operated within the sequence of events embedded within the climate model. Forcing and calibrating hydrologic models with simulated climate variables remains a marginal practice, but is yet documented in literature [35].

*2.3. Domain, Data, and Modeling Setup*

Asynchronous objective functions are tested over three intermediate size catchments (515–633 km$^2$) located in the St. Lawrence River Valley, Southern Quebec, Canada (Figure 3). Catchments are characterized by a nivo-pluvial hydrologic regime and a moderate slopes (5.9–6.4%). Forest is the dominant land cover type (59–77%) along with regenerating forest, wetlands and agriculture. The total annual precipitation is roughly 1000 mm while mean air temperature varies from −12 °C in January to 18 °C in July. Daily precipitation and temperature observations are interpolated by kriging to 0.1 degree, from in situ observations. Daily streamflow observations are extracted from hydrometric stations 022507 (Du Loup, 47.61° N, −69.64° E), 030101 (Nicolet Sud-Ouest, 45.80° N, −72.00° E) and 052233 (De l'Achigan, 45.90° N, −73.50° E).

The physically-based distributed hydrologic model WaSiM-ETH [36,37] was implemented over Du Loup, Nicolet Sud-Ouest and De l'Achiguan catchments (Figure 3, further details on the modeling setup are provided by Ricard and Anctil [13]). The river network is generated from a burned 50-m resolution digital elevation model (Figure 4a, Du Loup catchment is given as an example), resampled to 500 m and manually corrected. Land use is extracted from various sources provided by local agencies (Figure 4b). Percentages of clay, silt, and sand (Figure 4c) were retrieved from soil textures defined by Shangguan et al. [38].

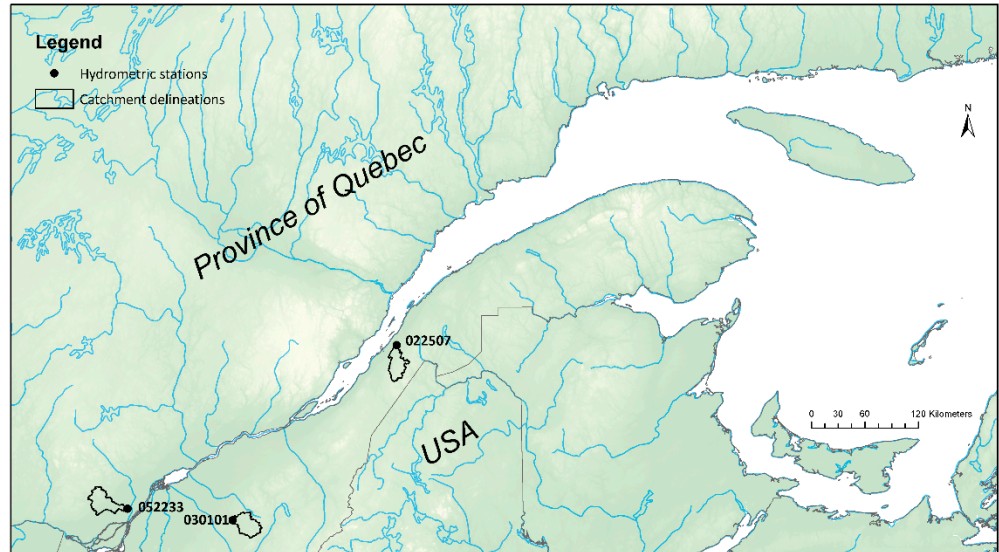

**Figure 3.** Du Loup (022507), Nicolet Sud-Ouest (030101) and De l'Achigan (052233) catchments.

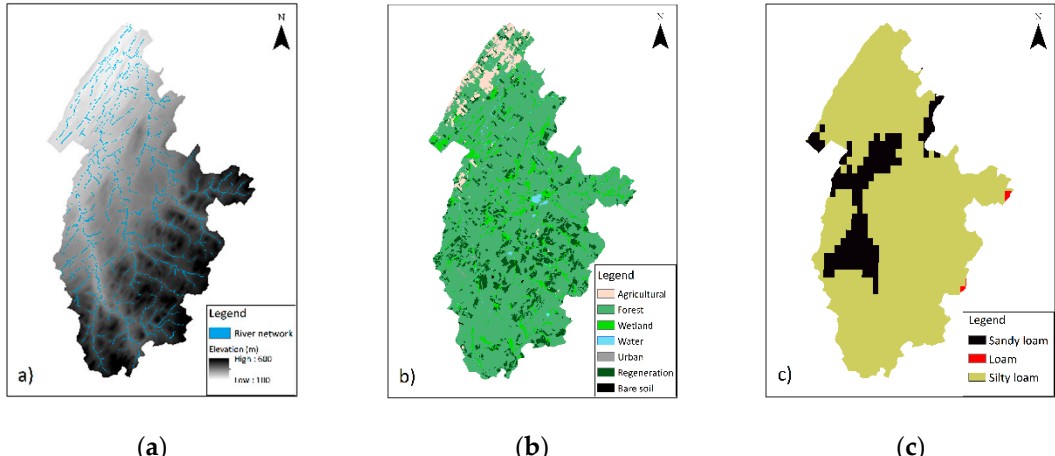

(**a**)　　　　　　　(**b**)　　　　　　　(**c**)

**Figure 4.** Du Loup River catchment (515 km$^2$). (**a**) Topography and river network, (**b**) land uses, and (**c**) soil textures.

Reference evapotranspiration ($E_0$) is evaluated using the Hamon temperature-based empirical formulation [39]:

$$E_0 = 0.1651 \cdot f_i \cdot \frac{h_d}{12} \cdot \frac{216.6 \cdot e_s}{T + 273.3} \tag{6}$$

where $f_i$ is an empirical correction factor (-), $h_d$ is the day length (h), and $e_s$ is the saturation vapor pressure at temperature $T$ (hPa).

Snowmelt is simulated using a temperature-index degree-day method [36]:

$$M = c_0(T - T_m) \cdot \frac{\Delta t}{24} \tag{7}$$

where $M$ is the melting rate (mm·d$^{-1}$), $c_0$ is a temperature-dependent melt factor (mm·°C$^{-1}$·d$^{-1}$), $T_m$ is the temperature limit for snow melt (°C), and $\Delta t$ is the time step (h).

Vertical fluxes within the unsaturated zone are based on Richards equation [40] applied to a 10-m deep column composed of 30 numeric layers. Surface runoff is a function of precipitation intensity

and hydraulic conductivity, while transient soil hydraulic properties follow the Van Genuchten equations [41]. The fraction of snow melt taken as surface runoff ($QD_{snw}$) is defined empirically [36]:

$$Q_s = Q_{snw} \cdot QD_{snw} \tag{8}$$

where $Q_s$ is the surface runoff (mm) and $Q_{snw}$ is the snow melt (mm).

Interflow ($Q_{int}$) is generated at soil layer boundaries considering slope and hydraulic conductivity [38]:

$$Q_{int} = k_s(\theta_m) \cdot \Delta z \; \cdot \; d_r \tan(\beta) \tag{9}$$

where $k_s$, the saturated hydraulic conductivity (ms$^{-1}$), $\theta_m$ is the water content in layer *m* (-), $\Delta z$ is the layer thickness (m), $d_r$ is a scaling parameter to consider river density (m$^{-1}$), and $\beta$ is the local slope angle (°).

Both surface runoff and interflow are delayed using recession constants [36]:

$$Q_{s,i} = Q_{s,i-1} \cdot e^{-\Delta t/k_s} + Q_s \cdot \left(1 - e^{-\Delta t/k_s}\right) \tag{10}$$

$$Q_{h,i} = Q_{h,i-1} \cdot e^{-\Delta t/k_h} + Q_h \cdot \left(1 - e^{-\Delta t/k_h}\right) \tag{11}$$

where $Q_{s,i}$ and $Q_{h,i}$ are delayed surface runoff and interflow at time step *i* (mm), $Q_s$ and $Q_h$ are the surface runoff and interflow at time step *i* (mm), $\Delta t$ is the time step (h), and $k_s$ and $k_h$ are recession constants (h).

Calibration of the model is operated using the Pareto Archived Dynamically Dimensioned Search optimization algorithm (PA-DDS, [42]) applied to the eight free parameters described in Table 2. Calibration is operated from 1980 to 1989 with a 1500-iteration budget. Validation is computed using the 1990–2009 period. For all simulations, we allowed an additional year for burning the hydrologic model. AOFs are evaluated in Section 3, relative to a seasonal variation of the Kling–Gupta efficiency (KGE$_s$, [26]):

$$KGE_s = \left[ KGE_{DJFMAM}; KGE_{JJASON} \right] \tag{12}$$

$$KGE = 1 - \sqrt{(r-1)^2 + (\propto -1)^2 + (\beta - 1)^2} \tag{13}$$

where DJFMAM refers to the period from December to May and JJASON, June–November, *r* is the correlation coefficient between the observed and simulated values, $\alpha$ is the ratio between the standard deviations, and $\beta$ is the bias. All components, including KGE, target 1 as the best score.

Simulated climate variables are extracted from the Canadian Regional Climate Model Large Ensemble (CRCM5-LE, [43]). The latter consists in the dynamical downscaling of the 50-member CanESM2-Large ensemble [44] using the CRCM5 [45] at a 12-km resolution over Northeastern North America. Climate simulations run from 1950 to 2100 following RCP8.5. For the purpose of the present study, daily mean air temperature and total precipitation are taken from the first member. Univariate quantile mapping is applied to simulated precipitation and temperature with a 50-bin transfer function, a monthly sub-scaling, and a three-month moving window. Precipitation below 1 mm is excluded from the calculation of the transfer function in order to prevent the 'drizzle effect' [46]. An additive correction is applied to air temperature while a multiplicative correction is applied to precipitation.

**Table 2.** Calibration parameters.

| Module | Calibration Parameter | Description | Unit | Boundaries |
|---|---|---|---|---|
| Reference evapotranspiration | $f_i$ | Seasonal correction factors (DJFMAM, JJASON) | (-) | [0.5;2] |
| Snow accumulation and melt | $c_0$ | Temperature-dependent melt factor | $(mm \cdot °C^{-1} \cdot d^{-1})$ | [0;5] |
| | $T_m$ | Temperature limit for snow melt | (°C) | [−2;2] |
| Unsaturated zone fluxes | $QD_{snw}$ | Fraction of surface runoff on snow melt | (-) | [0;1] |
| | $d_r$ | Scaling parameter for river density | $(m^{-1})$ | [1;100] |
| | $k_s$ | Surface runoff recession constant | (h) | [1;100] |
| | $k_h$ | Interflow recession constant | (h) | [1;150] |

## 3. Results

In Section 3.1, the hydrologic model is forced with climate observations over three catchments (Figure 3). In this common setup, observed and simulated streamflow values are synchronized. It is thus expected that a calibration based on the $KGE_s$ should dominate the AOFs, the former includes a correlation component. In Section 3.2, hydrologic projections are constructed over the climatic reference period for a single catchment. Configurations of the modeling chain are constructed using $KGE_s$ and most performing AOFs. In this case, it is hypothesized that the AOFs would do better because of the lack of synchronicity between the observed streamflow and the simulated climate time series.

### 3.1. Hydrological Performance over the Historical Period

Figure 5 presents the Du Loup River interannual and annual hydrographs (1990, 1995, 2000, and 2005) in validation, simulated by the WaSiM-ETH model forced and calibrated with historical meteorological observations. Calibration is steered either with $KGE_s$ (Equation (12)) or with asynchronous objective functions *AOF1–AOF5* (Section 2.1). The interannual performance is expressed in terms of *RMSD* (Equation (1), *RMSD* = 0 in case of perfect agreement) and the annual performance, in terms of KGE (Equation (13), which target = 1). Results show that $KGE_s$ calibration provides an accurate representation of the interannual hydrograph ($RMSD_{hst,KGEs}$ = 1.84 m³/s), with annual performance (KGE) ranging from 0.64 to 0.77. The synchronism of the nival peak flows is accurately represented but amplitudes are generally underestimated (1995, 2000, and 2005). *AOF1* leads to a hydrologic performance comparable to $KGE_s$. Synchronicity of the interannual hydrograph is marginally improved ($RMSD_{hst,AOF1}$ = 1.30 m³/s). *AOF1* annual performance improves in most cases (1995, 2000 and 2005), ranging from 0.75 to 0.86, since *AOF1* tends to improve the synchronicity of simulated nival peak flows but to degrade flow variance during the pluvial season, relative to the $KGE_s$. *AOF2* offers a much poorer representation of the interannual hydrograph ($RMSD_{hst,AOF2}$ = 10.4 m³/s). Annual performance falls to 0.37 from 0.66 mostly because the simulated nival peak flows are systematically overestimated or out of phase, while the pluvial season variance is underestimated. *AOF3* also offers a poor representation of the interannual hydrograph ($RMSD_{hst,AOF3}$ = 4.38 m³/s). Annual performance ranges from 0.35 to 0.70 mostly because the simulated nival peak flows are underestimated, while the pluvial season variance is improved relative to *AOF1*. *AOF4* leads to a moderately degraded interannual hydrograph ($RMSD_{hst,AOF4}$ = 3.22 m³/s). The amplitude of the mean annual nival peak flow is slightly overestimated while its recession synchronicity, out of phase.

Some annual hydrographs are improvements over the KGE*s* (1995, 2000, and 2005) but not systematically (1990). Flow variance is more accurate than other objectives-functions during the pluvial season. *AOF5* allows marginal improvements of annual hydrograph over KGE*s* ($RMSD_{hst,AOF5}$ = 1.62 m$^3$/s). Its annual performance is also very similar to *AOF1*, ranging from 0.75 to 0.89. It generally provides a robust representation of nival peak flows in terms of amplitude, timing, and volume, but it underestimates the pluvial flow variance.

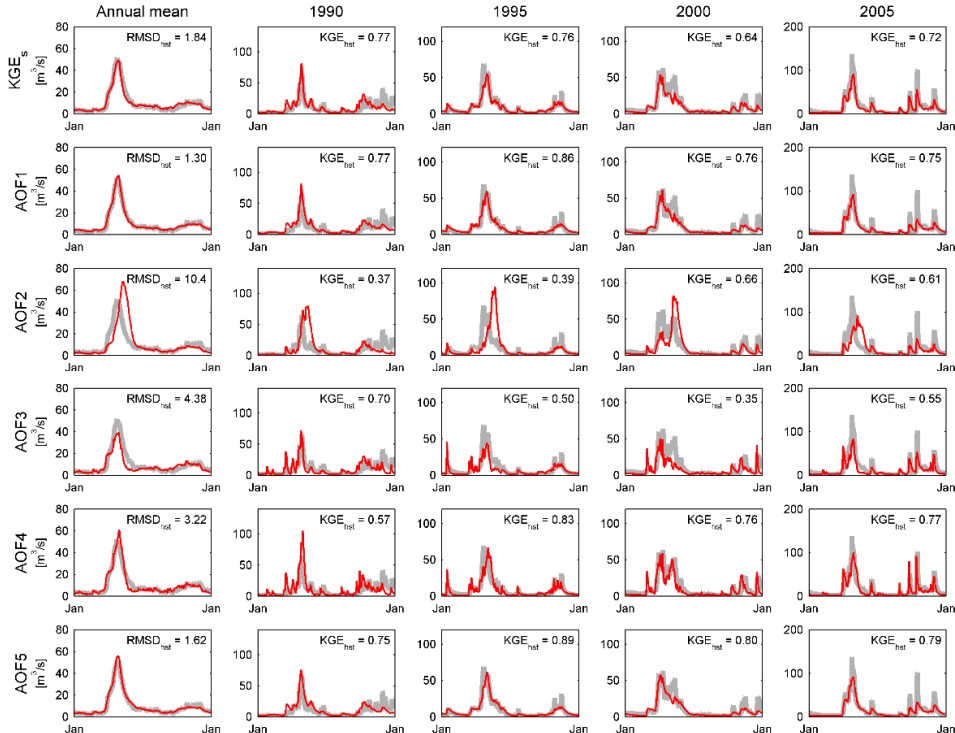

**Figure 5.** Observed (grey) and simulated (red) Du Loup River interannual and annual hydrographs (1990, 1995, 2000, and 2005, validation period). The WaSiM-ETH hydrologic model is forced and calibrated with historical meteorological observations (hst). Calibration is operated with a seasonal variation of the Kling-Gupta efficiency metric (KGE*s*) and asynchronous objective functions *AOF1* to *AOF5*. Interannual performance is expressed in RMSD, annual performance, in KGE.

Figure 6 illustrates the distribution of the validation annual performance values for calibration exploiting the KGE*s* or *AOF1* to *AOF5* for Du Loup, Nicolet Sud-Ouest and De l'Achigan catchments (Figure 3, sample size = 60, 20 years × 3 catchments). Performance is expressed in terms of KGE and its variance ($\alpha$), bias ($\beta$), and correlation (*r*) components (Equation (13)). Hydrologic performance is presented separately for the nival (DJFMAM) and pluvial (JJASON) seasons. Nival KGE*s* values range from 0.46 to 0.93: The median ($M_{DJFMAM}^{KGE,KGE_s}$) is 0.77. According to the non-parametric Wilcoxson rank-sum test, *AOF1* and *AOF5* offer nival performances comparable to the KGE*s* ($M_{DJFMAM}^{KGE,AOF1}$ = 0.73, $p$ = 0.16, $M_{DJFMAM}^{KGE,AOF5}$ = 0.73, $p$ = 0.21, significance level set to 0.05). *AOF4* do not lead to a comparable performance, the estimated *p*-value is however fairly close to the significance level ($M_{DJFMAM}^{KGE,AOF4}$ = 0.74, $p$ = 0.02). The poor *AOF2* or *AOF3* representation of the nival flow regime, depicted in Figure 5, is here confirmed. Their median annual performance values are significantly degraded relative to KGE*s* ($M_{DJFMAM}^{KGE,AOF2}$ = 0.56, $p$ = 2.59 × 10$^{-13}$ and $M_{DJFMAM}^{KGE,AOF3}$ = 0.55, $p$ = 1.13 × 10$^{-10}$). Poorer nival performances are driven by a severe degradation of the correlation (*r*), but also an overestimation of flow variance ($\alpha$) for *AOF2*. Most AOFs (not *AOF3*) improve nival bias ($\beta$) over the KGE*s*, while *AOF4* and *AOF5* improve flow variance. AOFs tend to degrade nival correlation relative to KGE*s*, the degradation remains moderate for *AOF1* and *AOF5*, as for *AOF4* to a certain extent. Hydrologic performance over the pluvial season

(JJASON) is generally poorer than for the nival season (DJFMAM) due to a degradation in both variance and bias. Pluvial performance of the KGE$_s$ ranges from $-0.22$ to $0.91$ with $M_{JJASON}^{KGE,KGE_s} = 0.64$. Most AOFs (not *AOF2*) present moderate but significant degradation pluvial performances relative to KGE$_s$ ($M_{JJASON}^{KGE,AOF1} = 0.53$, $p = 9.53 \times 10^{-5}$, $M_{JJASON}^{KGE,AOF2} = 0.37$, $p = 4.19 \times 10^{-6}$, $M_{JJASON}^{KGE,AOF3} = 0.57$, $p = 0.022$, $M_{JJASON}^{KGE,AOF4} = 0.56$, $p = 0.008$, $M_{JJASON}^{KGE,AOF5} = 0.55$, $p = 0.007$). The reduced AOF performance is driven by an underestimation of the flow variance (except for *AOF4*) and a degradation of the correlation. Most AOFs (not *AOF2*) improve however the pluvial bias representation ($\beta$) over the KGE$_s$.

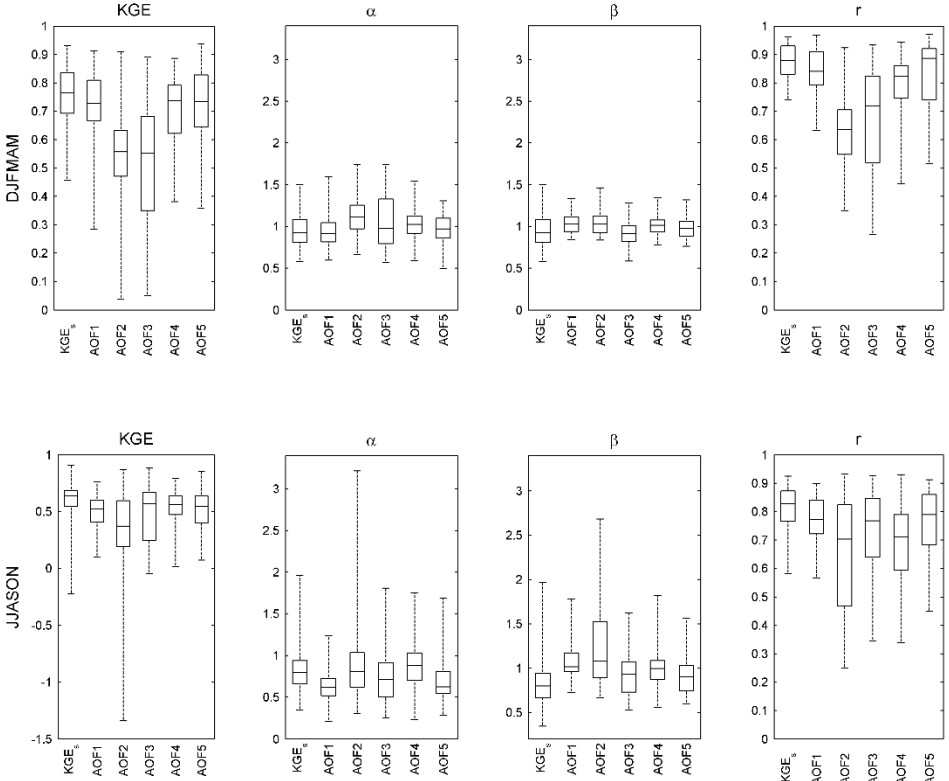

**Figure 6.** Hydrologic annual performance over the validation period for Du Loup, Nicolet Sud-Ouest and De l'Achigan catchments ($n = 60$). Calibration is operated with a seasonal variation of the Kling–Gupta efficiency (KGE$_s$) metric and asynchronous objective functions *AOF1* to *AOF5*. Performance is expressed in terms of KGE metric and its variance ($\alpha$), bias ($\beta$), and correlation ($r$) components from December to May (DJFMAM) and June to November (JJASON).

## 3.2. Hydrologic Projections over the Reference Period

AOFs present site-specific hydrological responses over the historical period (see Appendix B). By and large however, most performing AOFs are considered to provide fairly comparable behaviors from one catchment to another and further analyses of the hydrologic projections conducted in this section are limited to *Du Loup* catchment (Figure 3). Figure 7 presents raw and corrected interannual air temperature and total precipitation taken from the first member of CRCM5-LE as well as the local observations. As denoted by Leduc et al. (2019) over Northeast America, a strong warm bias reaching $+4\,°C$ affects winter air temperature (Figure 7a). Another $+2\,°C$ bias is observed in summer. Applying quantile mapping (Section 2.3) narrows notably the seasonal biases, except for a residual $\sim +1°C$ bias in January. This residual winter bias affecting temperature can be explained by the coarse monthly sub-scaling of the transfers function combined to the length of the three-month moving window, which is potentially inappropriate for such large seasonal biases. Leduc et al. (2019) also denoted a $+1$ to $+2$ mm/day bias affecting CRCM5-LE simulated precipitation. The impact of bias correction on simulated total annual precipitation can be observed in Figure 7b. A residual negative bias up to $-70$ mm/a exists

for precipitation totals above 1000 mm/a. The latter can be explained by the scaling mismatch between simulated precipitation and interpolated data from in situ observation or the application of a fixed threshold correcting the drizzle effect [47]. Nonetheless, in both instances, quantile mapping largely improves the climate simulation over the reference period.

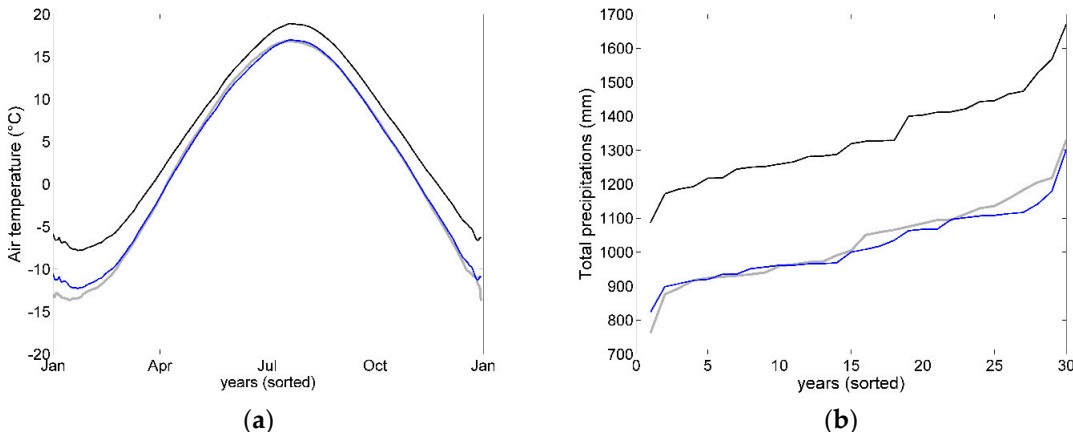

**Figure 7.** Observed (grey), raw (black), and bias-corrected (blue) interannual air temperature (**a**) and cumulative precipitations (**b**) simulated by the CRCM5-RCM over the Du Loup catchment.

Figure 8 compares the hydrologic projections simulated using both the conventional and alternative configurations of the hydroclimatic modeling chain (Figures 1 and 2), forced with bias-corrected air temperature and precipitation taken from the first member of the CRCM5-LE over the reference period (Figure 7). The WaSiM-ETH model is calibrated independently with $KGE_s$, *AOF1*, *AOF4*, and *AFO5* objective functions. *AOF2* and *AOF3* are excluded because their poor performance over the historical period (Section 3.1). Results are presented in Figure 8 as mean annual hydrographs and sorted logarithmic streamflow values below 10th and above 90th percentiles. Hydrologic projections are illustrated for both nival (DJFMAM) and pluvial (JJASON) seasons.

In all cases, projected mean annual hydrographs are affected by a notable degradation when compare to the performance achieved over the historical period ($RMSD_{hst}$ ranges from 1.30 m³/s to 1.84 m³/s, Figure 8). The amplitude of the projected nival flows is generally underestimated and out of phase. Within the conventional configuration (cnv), $KGE_s$ offers a weak representation of the mean interannual hydrograph ($RMSD_{cvn,KGEs} = 5.60$ m³/s). It also offers a systematic underestimation of peak flows combined to an overestimation of low flow, regardless the season. Mean annual hydrographs simulated with the alternative configuration (alt) are systematically better than for the conventional configuration. Integrating AOFs, $RMSD_{alt}$ ranges from 3.74 m³/s to 4.46 m³/s. The latter enhances the amplitude and timing of the projected nival mean flows, which translate into an average improvement in terms of $RMSD$ of 1.54 m³/s relative to $KGE_s$ within the conventional configuration.

The alternative configuration of the hydroclimatic modeling chain generally improves the representation of seasonal extremes values relative to $KGE_s$. *AOF5* provides the best representations of the projected nival peak flows ($RMSD_{alt,AOF5} = 0.03$), while *AOF1* and *AOF4* underestimate and overestimate the latter respectively ($RMSD_{alt,AOF1} = 0.07$, $RMSD_{alt,AOF4} = 0.08$). The representation of extreme nival low flows is more accurate using *AOF4* and *AOF5* ($RMSD_{alt,AOF4} = 0.10$, $RMSD_{alt,AOF5} = 0.09$), but notably overestimated using *AOF1* ($RMSD_{alt,AOF1} = 0.57$). Projected pluvial high flows are systematically underestimated. *AOF4* provides however a better representation than *AOF1* and *AOF5* ($RMSD_{alt,AOF4} = 0.22$, $RMSD_{alt,AOF1} = 0.47$, $RMSD_{alt,AOF5} = 0.43$). On the other hand, projected pluvial low flows are generally overestimated: *AOF4* and *AOF5* ($RMSD_{alt,AOF4} = 0.21$, $RMSD_{alt,AOF5} = 0.10$) providing a better representation than *AOF1* ($RMSD_{alt,AOF1} = 0.49$). Within the alternative configuration, $KGE_s$ provides a surprisingly accurate representation of the mean annual hydrograph ($RMSD_{alt,KGEs} = 4.42$ m³/s) but fail to translate into a good representation of seasonal extreme values.

Projected nival low flows and high flows and pluvial low flows are notably degraded relative to the conventional configuration.

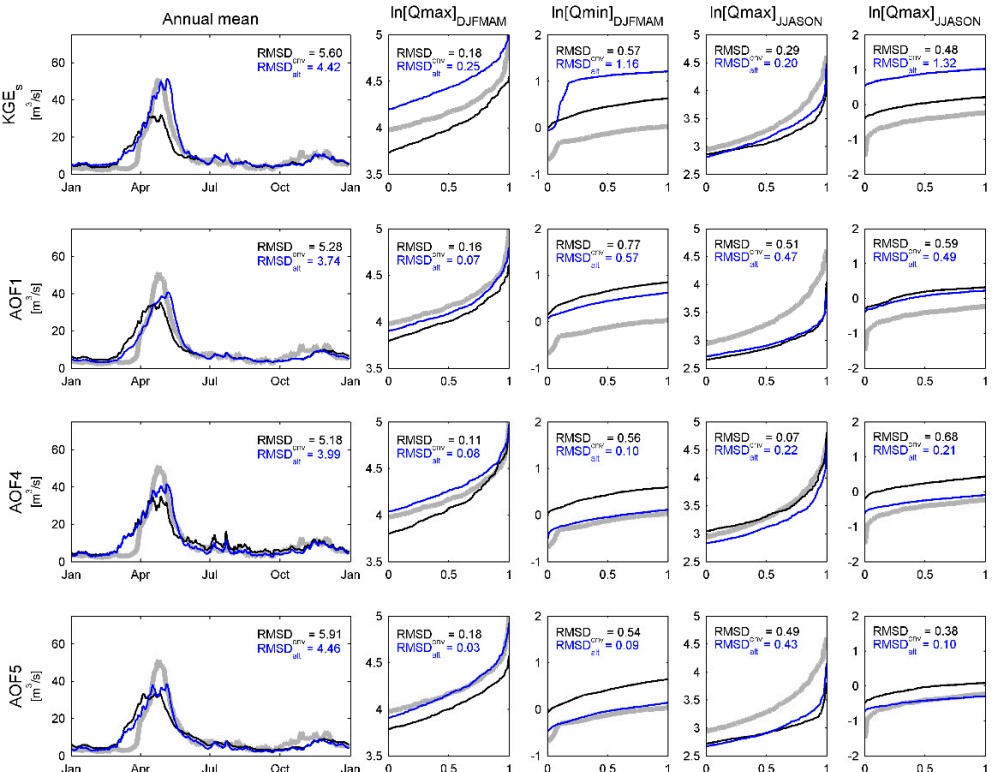

**Figure 8.** Observed (grey) and projected interannual hydrographs simulated by conventional (cvn, black) and alternative (alt, blue) configurations of the hydroclimatic modeling chain. The WaSiM-ETH hydrologic model is calibrated independently with $KGE_s$, *AOF1*, *AOF4*, and *AOF5* objective functions. Projections are also expressed through mean annual hydrographs and sorted logarithmic (ln) streamflow values below 10th and above 90th percentiles for both nival (DJFMAM) and pluvial (JJASON) seasons.

## 4. Discussion

### 4.1. AOFs *Provide an Appropriate Hydrological Response over the Historical Period*

Five exploratory asynchronous objective functions (AOFs) were designed and defined in Section 2.1 and used as calibration criteria purposely ignoring the correlation between simulated and observed streamflow. This property makes AOFs particularly suitable in a context where synchronicity between climate and hydrologic variables is unavailable; as in the case of constructing hydrologic projections using hydroclimatic modeling chains. Hydrologic performance of AOFs over the historical period was evaluated and compared to a seasonal variation of the Kling–Gupta efficiency metric (KGE). Results presented in Section 3.1 revealed the capacity of most performing AOFs to provide a hydrologic response comparable to KGE during the nival season, the latter being moderately degraded during the pluvial season. Hydrologic performance of AOFs is highly conditional to the application of a calibration criteria constraining an accurate representation of the interannual hydrologic cycle. AOFs excluding such a seasonal constrain (*AOF2* and *AOF3*) led to deficient representations of the flow regime. AOFs tend to reduce biases over both nival and pluvial seasons relative to KGE. Most performing AOFs also demonstrated a certain capacity to preserve the correlation, suggesting that the latter can be surrogated during the nival season by an appropriate interannual criteria within an AOF. AOFs tend to underestimate pluvial flow variance relative to $KGE_s$ except for *AOF4* that fairs better. *AOF3* finally provided a hydrological response comparable to other performing AOFs in a strictly pluvial regime.

### 4.2. The Alternative Configuration of the Modeling Chain Enforces the Consistency of Hydrological Projections

An alternative configuration of the hydroclimatic modeling chain was tested and compared to a conventional configuration in Section 3.2. The WaSiM-ETH hydrologic model has been forced over the Du Loup catchment with bias corrected climate variables taken from CRCM5-LE over the reference period, instead of historical meteorological observations. Both conventional and alternative configurations were constructed independently with $KGE_s$ and the best performing AOFs identified in Section 3.1 (*AOF1, AOF4*, and *AOF5*). The alternative configuration of the hydroclimatic modeling chain generally provided more coherent hydrologic projections relative to the conventional configuration. Projection of the interannual hydrograph was systematically improved, while seasonal extremes values were improved in most cases. The projected hydrologic response varied from one AOF to another. *AOF1* provided accurate projections of the mean annual hydrograph and nival peak flows. On the other hand, it underestimated pluvial peak flows and overestimated low flows, both nival and pluvial. The first order nature of *AOF1* (*RMSD* applied to interannual values) could explain its propensity to favor a sound representation of nival high flows to the expense of other hydrologic processes. In contrast, *AOF5* presented the most degraded projection of the interannual hydrograph, but the most performant projections of seasonal extremes values. Relative to *AOF1*, the construction by quantiles of *AOF5* weighted more effectively the extreme events. The absence of temporal sub-scaling within *AOF5* construction could explain the resulting degradation of the projected interannual hydrograph. Regardless a poorer performance over the historical period, *AOF4* presented a balanced projected hydrological response. Relative to other AOFs, *AOF4* did not notably degrade either the projected interannual hydrograph or seasonal extremes. Consequently, the application of a given AOF should be motived by the objectives defined in the scope of a given study. *AOF1* would be recommendable for assessment on water availability, *AOF5* for assessments on extremes, and *AOF4* for ecosystemic studies integrating multiple hydrologic considerations. Projections of the interannual hydrograph were affected by a systematic degradation relative to the historical period. This loss can be explained by a sensitivity of the hydrologic model to residual biases or inconsistency within post-processed climate variables. It can also be explained by a statistical mismatch between observed and simulated climate variable over the calibration period due to non-stationarity. Exploring a larger parametric space (Table 2) could have potentially provided better projected hydrological response, but would have further exposed the latter to overfitting.

### 4.3. Limitations

The work described in this manuscript is limited to many "singles": Single regional climate model, single member, single hydrologic model, single calibration period, etc. Generalising the applicably of AOFs within alternative configurations of the hydroclimatic modeling chain appears mandatory in further works. Applying the alternative configuration to a regional scale or large multi-members climate ensemble appears noteworthy challenges. Assessing the representability between calibration and validation periods considering non-stationary conditions between observations and multi-member climate simulations is another relevant question to further explore. Assessing the representability of the modelled sequence of extreme events prior to assessing formal statistical evaluation between the future and reference also appear relevant. Peaks-over-threshold statistical assessment could theoretically be very sensitive to a mismatch between the observed and modelled amount of extreme events over a given period. Other AOF constructions could also be tested. The intention while designing AOFs was to understand better their nature and behavior. More complex or performing AOFs can undoubtedly be designed, integrating additional *n*-moments, quantiles, or temporal sub-scaling. If available, other hydrological variables such as evapotranspiration, snow cover, or soil water content could be optimised in a similar way to *AOF1*. Variables simulated by a hydrologic model forced with historical meteorological observation could also surrogate observations. More complex constructions of AOFs would however further expose optimisation to equifinality and would thus request either an additional computing budget or adapted calibration strategies.

## 5. Conclusions

We introduced and tested an alternative configuration to the common hydroclimatic modeling chain with the aim of reinforcing the consistency of hydrological projections and circumventing the redundant usage of climate observations. We introduced a new type of calibration criteria, namely asynchronous objective functions (AOFs), which purposely ignore correlation between observed and simulated variables. The suggested configuration forces the hydrologic model with bias-corrected climate variables, thus preserving the sequence of events imbedded within the climate models. Results demonstrated that performing AOFs provided a hydrologic response comparable to the KGE metric over the nival historical period. They also demonstrated the capacity of the alternative configuration of hydroclimatic modeling chain to enforce the consistency of the projected interannual hydrograph and seasonal extreme values relative to a conventional configuration. AOFs presented distinct, but complementary, hydrologic responses, advocating for an appropriate application of AOFs according to the objective of a given study. The work described in this manuscript remains a proof of concept that requests further investigation and generalization to larger climate simulation ensembles, additional validation sites, and other climate regimes. This suggests, however, an innovative and fairly simple method enforcing the confidence affecting the production of hydrological projections.

**Author Contributions:** Conceptualization: S.R. and F.A.; methodology: S.R., J.-D.S., and F.A.; software: S.R.; validation: J.D.S. and F.A.; formal analysis: S.R.; investigation: S.R.; resources: J.-D.S. and F.A.; data curation: S.R.; writing—original draft preparation: S.R.; writing—review and editing: J.-D.S. and F.A.; visualization: S.R.; supervision: F.A.; project administration: J.-D.S. and F.A.; funding acquisition: J.-D.S., S.R., and F.A.

**Funding:** This research was funded by the Mitacs Accelerate program for scholarship to S.R. The authors wish to acknowledge the contributions of Ouranos and the Québec Ministry of Forests, Wildlife and Parks (MFFP project #142332118).

**Acknowledgments:** The authors also wish to acknowledge Quebec Ministry of Environment and Fight Against Climate Change (MELCC) for interpolated meteorological data (precipitation and temperature), hydrometric data, digital representation of the river network and integrated description of land uses.

**Conflicts of Interest:** The authors declare no conflict of interest.

## Appendix A. Methodological Framework

Step 1. Testing AOFs

- Five exploratory AOFs are designed (*AOF1–AOF5*).
- The hydrologic model is forced with climate observations over three sites.
- Optimization is conducted with AOFs and KGE as cost-functions.
- Performance is evaluated between synchronized observed and simulated streamflow values using the KGE metric.
- AOFs presenting inappropriate performance are excluded from step 2.
- Site-to-site variability of the resulting simulated hydrological response is evaluated before conducting step 2 to one site.

Step 2. Comparing conventional and alternative configuration of the hydroclimatic modeling chain

- Conventional and alternative configurations of the hydroclimatic modeling chain are implemented. The hydrologic model is forced with climate simulation.
- Optimization is conducted with KGE and most performing AOFs as cost functions.
- Performance of both configurations is evaluated between asynchronized observed and projected streamflow values using metrics excluding correlation.

## Appendix B. Site-to-Site Variability of the Simulated Hydrological Response

Figures A1–A3 present the distribution of the validation annual performance for Du Loup, Nicolet Sud-Ouest and De l'Achigan catchments evaluated independently (equivalent to Figure 6).

AOFs present site-specific hydrological responses over the historical period. While *AOF1* and *AOF5* outperform other AOFs over Du Loup catchment (Figure A1), AOFs offer a much more volatile response over Nicolet Sud-Ouest during the pluvial season (Figure A2). Long tailed distributions observed over De l'Achigan (Figure A3) are finally affected by isolated outlying weak annual performance values.

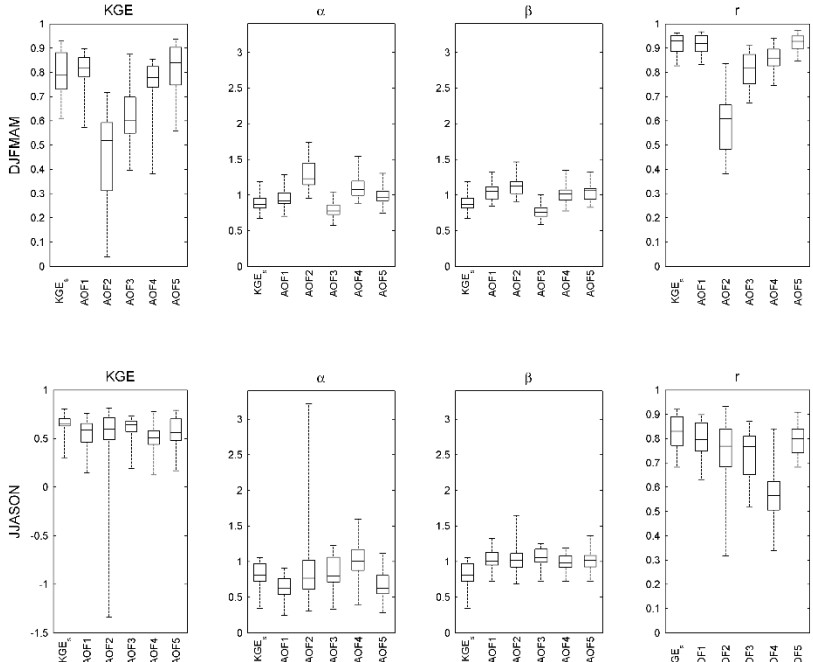

**Figure A1.** Hydrologic annual performance over the validation period for the Du Loup catchment (see Figure 6 for further description).

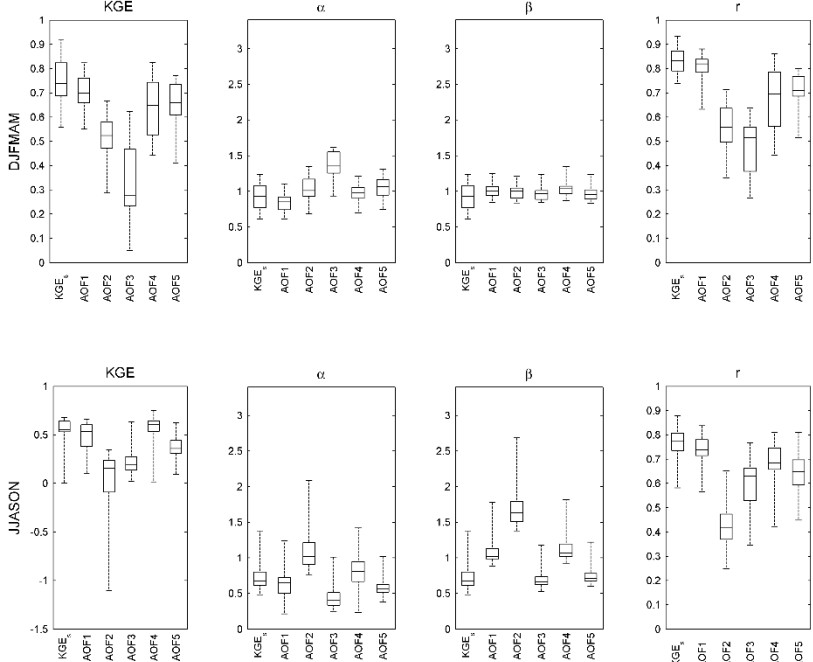

**Figure A2.** Hydrologic annual performance over the validation period for the Nicolet Sud-Ouest catchment (see Figure 6 for further description).

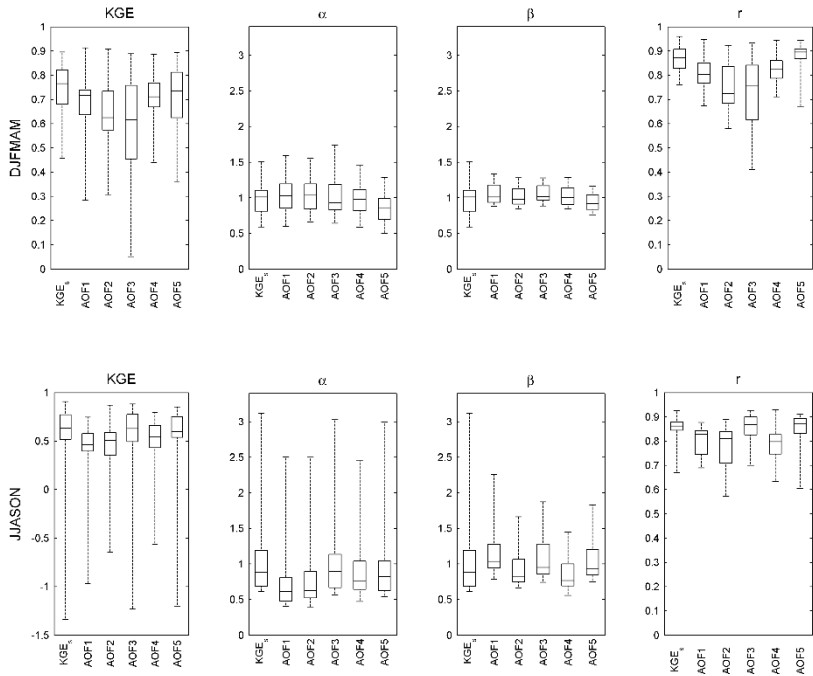

**Figure A3.** Hydrologic annual performance over the validation period for the De l'Achigan catchment (see Figure 6 for further description).

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
