# Peer review of "Exploring an Alternative Configuration of the Hydroclimatic Modeling Chain, Based on the Notion of Asynchronous Objective Functions"

_water, doi:10.3390/w11102012_

Round 1
Reviewer 1 Report
Review of "Exploring an alternative configuration of the hydroclimatic modeling change, based on the notion of asynchronous objective-functions" for Water (MDPI), Aug 2019
Summary
Ricard et al. present an alternative configuration of hydroclimate modeling using AOFs. This approach is applied to several small catchments in the Sanit-Lawrence Valley, Canada and compared to the Kling-Gupta efficiency metric. Forced with the observations AOFs provide a comparable response.
Overall the paper is well presented and written. Most of the figures are well constructed (though I do have some suggestions below) and the limitations somewhat addressed.
Two things to consider:
- The use of only one regional climate model is caveated, but wouldn't it be relatively simple to do this using output (that I assume is available) from several regional climate models? If so I would suggest that the authors do this, if the results remain the same this would vastly improve the contribution.
- Code, given that this is a methods based paper I was expecting that code would be provided for how the AOF constructions are carried out as well as and pre-processing, bias correction etc. Given the nature of this (methods) paper, and the fact that the authors state that they hope that this work will be generalizable in the future, I hope that code used to carry out the analysis (and produce Figures 5-7) would be made available.
Overall I am recommending minor revisions, my specific comments below will be easy for the authors to handle.
Specific Comments
Line 62: Is this figure based on one of the previous papers cited in the introduction? Based on the way it is described this seems to be the case.
Line 63: Change "manuscript" to "study" or just say "Here we propose..." (Same comment for line 74)
Line 107-109: Since the authors haven't introduced the study site it is a bit confusing to be defining the half year periods for analysis specifically by month. Suggest removing the parentheticals here and later defining the half year/biannual sub-scaling used by the authors (around paragraph starting line 134 for example).
Figures 1 and 2: I actually this it would be helpful if these two figures were combined into one, side by side. This would help identify the functional difference that the authors are suggesting here.
Line 137: If these are pristine forests this wouldn't be "land use", rather "land cover type" of something similar.
Figure 4a: This is hard to see the topography and river network. I would suggest a shaded relief image with clearer (crisper) stream network shown.
Lines 175-177: Need to define all variables in KGE equation in text.
Lines 154-186: Please ensure original citations to all of these equations are included here.
Line 191: "It is expected.." based on? Citation or example is perhaps required.
Line 202 and 203: "which target"? I believe the authors mean perfect agreement.
Figure 7. The axes are not clear on this figure and it is generally hard to read. Please improve this.
Line 280: Better to describe the "sorted total precipitations" as "cumulative precipitation".
Line 404: "further investigation and generalization" - I suggest the authors expand not his point. I agree, further work in other climate regimes and elsewhere in the world would be helpful.
Author Response
Response to Reviewer 1 Comments
Point 1. The use of only one regional climate model is caveated, but wouldn't it be relatively simple to do this using output (that I assume is available) from several regional climate models? If so I would suggest that the authors do this, if the results remain the same this would vastly improve the contribution.
Response 1. The reviewer is right, the use of many regional climate models (RCMs) would have improved the generalization of the study. The use of a single RCM is, however, highlighted as a limitation in lines 398 to 402 of the revised manuscript. Since the alternative configuration of the hydroclimatic modeling chain requires time-consuming iterative optimization, operating the latter with numerous RCMs cannot be realised within the deadlines given by the editor. We drove this study with the aim of building a proof of concept, introducing the innovative notion of asynchronous objective-function (AOF). We thus prioritized designing and testing multiple forms of AOF. Applying the alternative configuration to numerous RCMs will finally put upfront the contribution of parametric equifinality to the global uncertainty affecting the modeling chain. In our view, this question should be investigated within a distinct methodological framework and thus presented into another manuscript.
Point 2: Code, given that this is a methods based paper I was expecting that code would be provided for how the AOF constructions are carried out as well as and pre-processing, bias correction etc. Given the nature of this (methods) paper, and the fact that the authors state that they hope that this work will be generalizable in the future, I hope that code used to carry out the analysis (and produce Figures 5-7) would be made available.
Response 2: Most codes used for this study (quantile mapping, Pareto-based optimization, KGE metrics) are already available and shared among the community. We will be pleased to share codes computing AOFs.
Specific Comments
Point 3: Line 62: Is this figure based on one of the previous papers cited in the introduction? Based on the way it is described this seems to be the case.
Response 3: This figure is not based nor adapted from the previous papers cited in the introduction.
Point 4: Line 63: Change "manuscript" to "study" or just say "Here we propose..." (Same comment for line 74)
Response 4: “Manuscript” has been changed to “study” in lines 71 and 82.
Point 5: Line 107-109: Since the authors haven't introduced the study site it is a bit confusing to be defining the half year periods for analysis specifically by month. Suggest removing the parentheticals here and later defining the half year/biannual sub-scaling used by the authors (around paragraph starting line 134 for example).
Response 5: We acknowledge describing the sites before the methodological framework would have eased the reading of the manuscript. In order to minimize editing of the latter, we added a reference to section 2.3 at line 134.
Point 6: Figures 1 and 2: I actually this it would be helpful if these two figures were combined into one, side by side. This would help identify the functional difference that the authors are suggesting here.
Response 6: Figure 2 have been moved to the introduction. We believed this will ease the comparison of conventional and alternative configurations.
Point 7: Line 137: If these are pristine forests this wouldn't be "land use", rather "land cover type" of something similar.
Response 7: ‘land use’ had been changed to ‘land cover type’ in line 159.
Point 8: Figure 4a: This is hard to see the topography and river network. I would suggest a shaded relief image with clearer (crisper) stream network shown.
Response 8: Figure 4a has been edited according to the reviewer’s comment.
Point 9: Lines 175-177: Need to define all variables in KGE equation in text.
Response 9: A description of all variables is added in lines 201 to 203. Thank you.
Point 10: Lines 154-186: Please ensure original citations to all of these equations are included here.
Response 10: Original citations to equations (6) to (11) are included. References are edited consequently.
Point 11: Line 191: "It is expected." based on? Citation or example is perhaps required.
Response 11: Clarifications are provided in lines 216-218.
Point 12: Line 202 and 203: "which target"? I believe the authors mean perfect agreement.
Response 12: Indeed. Text edited consequently at line 228.
Point 13: Figure 7. The axes are not clear on this figure and it is generally hard to read. Please improve this.
Response 13: Axes in Figure 7 have been improved.
Point 14: Line 280: Better to describe the "sorted total precipitations" as "cumulative precipitation".
Response 14: "Sorted total precipitations" has been changed to "cumulative precipitation" in line 306.
Point 15: Line 404: "further investigation and generalization" - I suggest the authors expand not his point. I agree, further work in other climate regimes and elsewhere in the world would be helpful.
Response 15: The "further investigation and generalization" point is expanded to climate ensemble, validation sites and other climate regimes at lines 430 and 431.

Reviewer 2 Report
The concept of the manuscript is new and innovative. As a reader, I felt like the proposed methodology is not supported by enough scientific evidences to be accepted.
The introduction part of the manuscript is weak and does not provide enough references as foundations or explanations to the newly proposed hypothesis which goes against traditional standards set by hydrological sciences community.
The manuscript generates confusion in many areas. For instance, the verification and validation strategy is often not clear and highly inconsistent. If two methodology (herein traditional hydrological modeling chain and proposed one) needs to be compared fairly, each strategy should be validated across a consistent time period and river basins using the same set of metrics. This is not the case here and it seems like the authors picked and chose only those results that support the hypothesis of the manuscript which is against standard scientific practice. As such, I believe the manuscript needs to be improved in many areas before it can be considered for publication.
Author Response
Response to Reviewer 2 Comments
Point 1. The introduction part of the manuscript is weak and does not provide enough references as foundations or explanations to the newly proposed hypothesis which goes against traditional standards set by hydrological sciences community.
Response 1. We understand the comment of the reviewer. The redundant use of climate observations implementing hydroclimatic modeling chain is a widely common practice. To our knowledge, its contribution to the global uncertainty affecting hydrologic projections is not documented in literature. Based on additional references, we clarify (lines 50 to 60 of the revised manuscript) the purpose of the proposed configuration and related hypotheses. We believe these lines improves the introduction of the manuscript.
Point 2: The manuscript generates confusion in many areas. For instance, the verification and validation strategy is often not clear and highly inconsistent. If two methodology (herein traditional hydrological modeling chain and proposed one) needs to be compared fairly, each strategy should be validated across a consistent time period and river basins using the same set of metrics. This is not the case here and it seems like the authors picked and chose only those results that support the hypothesis of the manuscript which is against standard scientific practice. As such, I believe the manuscript needs to be improved in many areas before it can be considered for publication.
Response 2: The proposed methodological framework resorts to many novel steps. Lines 91 to 101 and Appendix A (lines 444 to 459) brings additional information describing to the step-by-step methodological framework. We believe these latter brings clarity and enforce the justification of the research design. We also believe the arguments presented below support the fact that our conclusions are supported by the result presented within the manuscript.
The first step aims at testing five exploratory AOFs in a standard, common modeling framework (hereafter referred to as the “historical modeling framework”): the hydrologic model is forced with climate observations and performance is evaluated between synchronised observed and simulated streamflow values. AOFs are compared to the KGE metric from December to May (nival season) and from June to November (pluvial season) over three catchments using a state-of-art split-sample test. AOF2 and AOF3 are excluded from further analysis because they offer a lesser performance than the others in this setup. We consider this historical modeling framework a necessary condition before an AOF could be used for the construction of an alternative configuration to the hydroclimatic modeling chain.
We also analysed the site-to-site variability of the simulated hydrological response. The latter was considered sufficiently identical to conduct further analysis (hydrological projections) at a single site. We acknowledge this decision is debatable, our main motivation was to provide clear and concise results to analyse. Hydrological projections can easily be produced and analysed over other sites, but not within the deadlines provided by the editor. It is our belief, however, that we would end up with strictly the same findings.
Producing hydrological projections from climate model simulations, observed and resulting projected streamflow values are not synchronised by construction. Evaluating the performance of hydrological projections, we decided not to use the KGE metric but rather statistical criteria excluding correlation. The evaluation of the split sample-sample test (calibration/validation periods) remains consistent with the previous analyses. Within this framework and considering limitations discussed in the manuscript, we believe alternative and conventional configurations are fairly compared.

Round 2
Reviewer 2 Report
I recommend the manuscript can now gets accepted at its current form for publication. Thanks.